# Mono-Dispersed Microspheres Locally Assembled on Porous Substrates Formed through a Microemulsion Approach

**DOI:** 10.3390/polym12040964

**Published:** 2020-04-21

**Authors:** Jianfeng Zhang, Shuxin Gong, Jiahang Zhu, Jiejing Zhang, Jing Liang

**Affiliations:** College of life science, Key Laboratory of Straw Biology and Utilization, the Ministry of Education, Jilin Agricultural University, Changchun 130118, China; zhangjianfeng06@tsinghua.org.cn (J.Z.); gsx1127117149@163.com (S.G.); zhujiahangzhujia@163.com (J.Z.); zjjx124@163.com (J.Z.)

**Keywords:** microemulsion, porous structures, mono-dispersed microsphere array

## Abstract

A cost-effective, simple, and time-saving method to fabricate mono-dispersed periodic microsphere structures on substrates with patterned sites is very meaningful due to their significance on various biological studies. Herein, a simple and facile method to fabricate mono-dispersed microsphere arrays on porous substrates was developed. The mixture of polystyrene and an organic stabilizer solution which contains aqueous solution, fabricated through shaking, was applied to prepare microemulsion solution. An ordered porous structure was produced by spreading and evaporating the solvent of microemulsion on a glass slide, accompanied by the enrichment of didodecylamine in the cavities. The porous cavities were further modified with polyacrylic acid and poly(diallyldimethylammonium chloride) which could immobilize the microspheres. The charged microspheres were incorporated into the cavities by an electrostatic interaction with the oppositely charged polyelectrolytes. The positive polyelectrolytes with abundant charges as well as a suitable content and dimension of microspheres, ensured the formation of mono-dispersed and ordered arrays. Considering that other charged particles were universally suitable for the present strategy, the reported approach opened an efficient way for the preparation of microsphere-based materials.

## 1. Introduction

Mono-dispersed microspheres have attracted wide attention owing to their excellent features of large surface areas, high diffusibility and mobility, uniformity, and variety [1,2]. The periodic patterns from mono-dispersed spheres have been applied on the fields of porous materials, photonic crystals, and so forth [3,4,5]. Methods have been developed for the preparation of an ordered pattern consisting of colloid particles, such as photolithography, self-assembly, inversed opals, and so forth [6,7,8]. In addition, most microsphere patterns are prepared on planar substrates, which are involved in elaborate and complicated processes, expensive instruments, as well as long time and professional operators, such as in photolithography techniques, and require limited materials, for example, self-assembly and inversed opals approaches. Thus, the formation of mono-dispersed periodic microsphere structures, which are important for various biological studies, through a cost-effective, simple, and time-saving method, is very meaningful.

Microsphere-based materials are famous in the field of immunoassays, nano-imaging, sensor, catalysis, and so on, due to their large reactive and easily modified surfaces [9,10,11,12]. However, the methods to prepare microsphere-based materials in a random arrangement are quite expensive and complicated [13]. Thus, low-cost strategies to fabricate ordered microspheres on substrates with patterned structures are necessary. 

Breath figure is a famous self-assembly method to prepare porous surfaces, which applies water droplets as template, has been widely investigated, due to its fast, easy, and cheap features [14,15,16,17]. A series of colloid particles and microspheres, for example, hemispherical TiO_2_ microparticles, silica particles, and polymer microspheres, have been assembled into the cavities to construct particle-based materials [18,19,20]. However, for the assembly of polymer spheres onto the patterned substrates by the breath figure method, the current strategy either involves complicated procedures such as the treatment of patterned substrates by UV-ozone to improve the surface wettability to deposit the polymer microspheres [21,22], or refers to poly-dispersion of polymer spheres on the patterned substrates [20]. Herein, mono-dispersed microspheres on porous substrates, where the microspheres were incorporated into the cavities and the porous film acts as the substrates, have been fabricated through a modified breath figure method called microemulsion approach (Scheme 1). This strategy shows several advantages. The microspheres are assembled on the porous substrates in a direct, cheap, facilitated, and site-specific method without complicated pre-treatment steps. The microspheres are in excellent mono-dispersed arrangement with a high order. Interestingly, the large number charges of polyelectrolytes and the suitable content and dimension of microspheres ensure a fabrication of the mono-dispersed and ordered arrays. The mono-dispersed microsphere arrays may provide a novel platform for biological research, such as immunoassays and pattern recognition.

## 2. Materials and Methods

### 2.1. Materials

Polystyrene (PS, molecular weight: 349 kg/mol), polyacrylic acid (PAA), poly(diallyldimethylammonium chloride) (PDDA), and dichlorotris(1,10-phenanthroline)ruthenium(II) hydrate (CAR) were obtained from Sigma-Aldrich (Shanghai, China), respectively. Didodecylamine (DDA) was purchased from Aladdin (Shanghai, China). The carboxyl polystyrene microspheres (2.5% *w*/*v*), suspended in deionized water and fluorescein disodium salt (FDS), were acquired from J&K Scientific Ltd. (Beijing, China). 

### 2.2. Preparation of Porous Film

The mixture of PS (6 mg/mL) and DDA (0.3 mg/mL) was prepared by the addition of them into dichloromethane. For the preparation of a microemulsion solution, the typical procedure was to add a certain volume fraction of water maintained at 5% to the mixed solution of PS and DDA. Then, the mixture solution was shaken for 30 s to achieve a translucent gray microemulsion. After that, the microemulsion solution with about 20 µL was cast onto a glass slide under condition of the relative humidity of 30–40% and temperature of 25 °C to obtain PS porous films. 

### 2.3. Assembly of PAA and PDDA into the Porous Cavities

For the incorporation of PAA into the porous cavities, PS porous film was dipped into 1 wt % PAA aqueous solution for 30 min, followed by washing with water three times and then drying in air. For the detection of the PAA assembled into the cavities, the PS porous film was immersed into a 0.5 mM aqueous solution of CAR dye for 30 min, followed by washing with water three times and drying in air. For the assembly of PDDA into the cavities, PS film with PAA was dipped into 1 wt % PDDA aqueous solution for 30 min, followed by washing with water three times and drying in air. For the varication of PDDA added into the cavities, the modified film was dipped into a 0.5 mM aqueous solution of FDS for 30 min, followed by washing with water three times and drying in air.

### 2.4. The Assembly of PS Microspheres into the Cavities

The obtained PS microsphere suspending solution was diluted to 0.1% *w*/*v* by adding deionized water. Then, the modified PS film was immersed into the PS solution for 3 h followed by shaking the solution for 30 s every 15 min. After that, the PS-contained film was washed with water three times and dried in air.

### 2.5. Measurements

Scanning electron microscopy (SEM) images were collected on a JEOL JSM-6700F field emission scanning electron microscope (JEOL Ltd., Tokyo, Japan). Confocal laser scanning microscopic (CLSM) images were obtained using an Olympus FluoView FV1000 (Olympus Ltd, Tokyo, Japan). Analysis of CLSM data was carried out by using the FV10 ASW software (Olympus Ltd, Tokyo, Japan). Dynamic light scattering (DLS) was performed on a Zetasizer NanoZS (Malvern Instruments, Malvern, UK).

## 3. Results and Discussion

### 3.1. Preparation and Structural Characterization of Polymer Porous Films

The breath figure method, applying water droplets as template to fabricate porous film, was usually operated at a high humidity environment to guarantee the source of the sacrificial agent. Recently, a modified method named the microemulsion approach was developed, which only needs a mild condition as opposed to a high humidity atmosphere [23,24]. The microemulsion method provides a much easier route to decorate the inner walls of the cavities, leading to the assembly of various functional water–miscible components [25]. In this strategy, the microemulsion was prepared through mixing of PS organic solution containing DDA with water and following a slight shaking, which evenly dispersed the microemulsion droplets in the organic solution. After spreading the microemulsion onto a glass slide followed with the evaporation of the solvent, the porous polymer film was obtained, as shown in Scheme 1a. The process can be divided into several steps: firstly, the organic solution of polymers and the water are mixed by shaking to generate a reversed microemulsion solution, where the dispersed water droplets are in several hundreds of nanometers in the organic solution; secondly, after casting the emulsion solution on a solid substrate, the thermal motion of water droplets causes the inevitable collision and coalescence, making the size of those microemulsion droplets increase gradually; thirdly, driven by thermal capillary and convection, the water droplets start to arrange, forming hexagonal close-packing; finally, when the solvent and water evaporate completely, the water droplets leave an imprint on the polymer surface, forming the honeycomb pattern. The prepared polymer film shows bright iridescent colors when viewed in the reflection of light, indicating a periodic refractive index variation regarding the film thickness. The surface structure of the porous film was further characterized by SEM analysis. The SEM image in Figure 1a exhibits a highly ordered patterned structure with mono-dispersed hexagonal close-packed holes in a large scale without defect. The histogram (Figure 1) illustrates the size distribution of the cavities with the diameter between 3.3 to 3.8 µm, indicating the uniformity of the dimension of the cavities.

### 3.2. Selective Assembly of PAA and PDDA into Patterned Cavities

As we know, the amphiphilic additives stabilize water droplets during the preparation of a microemulsion solution and porous cavities, whose hydrophilic groups, such as amine of DDA, specifically locate at the inner surface of the cavities after their formation. In order to facilitate the assembly of charged microspheres, polyelectrolytes soluble in water taking large charges were employed. Due to the amine groups with positive charges from a water droplet stabilizer located in the cavities, oppositely charged PAA can be introduced into the cavities through electrostatic interaction. The assembly of PAA into the cavities was verified by CLSM and XPS analysis. To examine the incorporated situation, the PAA-contained film was bound with the cationic CAR dye molecules in the aqueous solution. As displayed in Figure 2a, a strong red hexagonal fluorescent array of CAR has red emission appearing at the sites of the cavities throughout the film. In a similar procedure, the PS porous film without PAA decoration was dipped into the CAR solution, no fluorescence was obtained, as shown in Figure 2b. These results demonstrated the successful building of a PAA layer in the cavities.

To provide more positive chargers for incorporating microspheres, a positively charged polyelectrolyte PDDA was employed. After pre-assembly of PAA into the cavities, PDDA, which bears ammonium groups, was then incorporated into the cavities through the electrostatic interaction between carboxylic acid and ammonium groups (Scheme 1). The PDDA-modified films were prepared by simply soaking PAA-contained film into PDDA solution. The assembly of PDDA was determined by CLSM measurement. After treatment of PAA-contained film with PDDA, it was subsequently dealt with FDS dye solution. The CLSM image in Figure 2c shows green domains in an ordered, hexagonal structure which is the same as that of the arrangement of cavities, indicating the formation of PDDA layers. In addition, the PAA-decorated film was also characterized by XPS. The peak of the oxygen atom supports that the PAA is introduced into the film due to the lack of oxygen elements in the original PS and DDA films (Figure 2d).

### 3.3. The Build of Microsphere-Based Materials

PS microspheres are usually a good choice due to their easily operated and modified features. In this strategy, microspheres with a suitable dimension comparable to the cavity diameters of the porous structures are in demand. The selected microspheres display a narrow size distribution. The average diameter of the microspheres is about 3 µm which is close to the dimension of the cavities, as shown in Figure 1 and Figure 3. 

Microsphere-based materials have potential on biological areas due to their large and reactive surface. To build microsphere-based materials, porous films with PAA and PDDA layers acting as substrates were immersed into the PS suspending solution. As the modification of PDDA occurs selectively in the cavities, they provide a large amount of positively charged units to bind with the negatively charged PS microspheres through electrostatic interaction. The patterning of the microspheres on the substrate was visualized using SEM measurement (Figure 4). All the microspheres selectively incorporated into the patterned cavities rather than on the top surface of the film with a suitable concentration of microspheres (Figure 4a,b). Interestingly, every cavity only accommodates one microsphere, indicating the perfect dimensional compatibility between the cavities and the microspheres. However, only several microspheres could be removed from the pores with washing steps as shown in Appendix A, indicating that the electrostatic interaction ensured the immobilization of microspheres onto the patterned sites. The assembled situation could be modulated in some extent through the alternation of the microsphere concentration. A lower content of microspheres leads to some empty cavities without microspheres (Figure 4a), but a higher content results in overflowing microspheres on the top surface with each cavity filled with one microsphere (Figure 4c). Thus, to prepare ordered mono-dispersed microsphere arrays, a proper concentration of PS microspheres with adequate but not excessive dimensions is necessary (Figure 4b). In addition, the charged layers with numerous charges at the inner surface of cavities is another crucial issue, and just a small number of charges with the assistance of gravity leads to a low filling ratio of the microspheres. As described in Wan’s research, unquaternized films with no charges exhibited a very low filling ratio of polymer spheres taking carboxyl groups [20]. However, quaternized films possessing positive charges showed a much better filling amount of the polymer spheres with carboxyl groups characterized through the SEM images, indicating that an electrostatic interaction between charged pores and oppositely charged spheres was responsible for the site-specific assembly. Finally, microspheres with a size similar with that of the cavities is also important, because a larger microsphere has no ability to enter the cavities, and a much smaller microsphere may lead to several microspheres in one cavity which will damage the mono-dispersed arrangement. Thus, a suitable content and a proper size of microspheres, as well as a large number of charges of substrates, have an important contribution in the formation of the mono-dispersed ordered microsphere-based materials, which could be used in biological aeras. 

## 4. Conclusions

In conclusion, the mono-dispersed microsphere-based materials, where the microspheres locate on the cavities and the porous film serves as substrate fabricated through a microemulsion method, have been developed. The process is much simpler, more facile, and site-specific in the fabrication of microsphere-based materials. Herein, the microspheres were locally assembled in the patterned sites on the film in several simple steps, saving some expensive costs compared to other methods. Interestingly, numerous positive charges of polyelectrolyte PDDA, proper concentration of microspheres, and dimension of microspheres similar with the size of the cavities, guarantee the order and mono-dispersed features of microsphere-based materials. It can be envisioned that other charged particles with diverse physical, chemical, or biological properties also are favorable for the preparation of microsphere-based materials, which further endow these types of materials with a variety of potential applications in photic, sensor, biomedicine, and so forth.

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
