# Peer review of "Mono-Dispersed Microspheres Locally Assembled on Porous Substrates Formed through a Microemulsion Approach"

_polymers, 2020, doi:10.3390/polym12040964_

Round 1
Reviewer 1 Report
The manuscript by Liang Jing and co-workers reports a simple protocol for assembly of PS microspheres on a PS substrate featuring ordered pores of uniform size which were additionally impregnated with polyacrylic acid and poly(diallyldimethylammonium chloride). It is quite interesting from the methodological point of view although presentation of some practical aspects of elaborated approach would be valuable. Despite this limitation, the work seems to be competently done and technically sound. It may stimulate further work focused on direct applications and thus, in my opinion, it deserves publication in Polymers. However, the manuscript shows a number of language deficiencies and I expect that it will be carefully checked and revised prior to final acceptance.
Author Response
Question 1: The manuscript shows a number of language deficiencies and I expect that it will be carefully checked and revised prior to final acceptance.
Reply: Thanks for the comment. We have carefully checked and polished the language. The corresponding discussion is marked as red font in the text.
Reviewer 2 Report
Author developed a method to fabricate microsphere based materials with mono-dispersed features in a simpler, more facile and low cost. Since site specific mono-dispersed microspehere are important in fields of porous materials and photonic crystals, this will attract significant attention to the readers. In this perspective, I recommend this work for publication.
Author Response
Thanks for the comment.
Reviewer 3 Report
Authors described the preparation of assembly of mono-dispersed microspheres on a porous substrate. The SEM images confirmed the mono-dispersity of spheres as well as their assembly. This work is interest of the field however, there are few points needs to be addressed before its publications.
1- In Scheme 1, authors summarize the process but they should describe also how porous substrate is made as well. This should be clearly described in the manuscript as well.
2- The method used here should be clearly distinguished from the literature methods in the introduction and merits/demerits should be discussed.
3- Authors mentioned a lot about how the surface charges and interactions between the spheres and pores are important holding this structure together but there is no spectroscopic characterization to support any of the claims. I understand if authors cannot perform further characterizations due to ongoing pandemic but at least they should acknowledge this fact and cite the appropriate literature to support their claim.
4- Can the microspheres be removed from the pores with washing? This should be discussed in the manuscript.
